# INTERACTIVECOT: ALIGNING DYNAMIC CHAIN-OF-THOUGHT PLANNING FOR EMBODIED DECISION-MAKING

## ABSTRACT

Vision-Language Models (VLMs) are increasingly being employed as the decision-making "brains" of embodied agents. Effectively harnessing their powerful generalization capabilities in dynamic, context-specific tasks remains a significant challenge. Chain-of-Thought (COT) prompting is often utilized for complex task execution, but existing methods either rely on static strategies that fail to adapt to changing environments or fine-tune on offline datasets, which are insufficient for optimizing agent decision-making through interaction. In this paper, we propose a novel approach that focuses on optimizing the COT reasoning process rather than just the final action tokens. By aligning the COT process through preference-based reinforcement learning, specifically Direct Preference Optimization (DPO), we enhance the agent's ability to make accurate decisions in dynamic environments while mitigating model degradation during fine-tuning. Our method models the environment as a Markov decision process, requiring the agent to reflect on the current state in real time to generate adaptive plans and actions. By prioritizing the optimization of the COT process over the final actions, we enhance the agent's reasoning adaptability while effectively mitigating model degradation during fine-tuning. Experiments in the ALFWorld environment demonstrate an average success rate of **26.67%**, which is a **6%** improvement over RL4VLM, and show that our method effectively mitigates model degradation post fine-tuning. These results highlight the potential of integrating preference-based reinforcement learning techniques with COT processes to enhance the decision-making capabilities of vision-language models in embodied agents.

## 1 INTRODUCTION

Large Language Models (LLMs) and Large Multimodal Models (LMMs) have achieved remarkable success in natural language understanding and generation tasks (Brown, 2020; Achiam et al., 2023). Recent studies have explored how LLMs can be leveraged to manage other AI models and tools for complex language or multimodal tasks (Shen et al., 2024; Lu et al., 2024), assist in playing sophisticated games such as TextWorld (Yao et al., 2022), Handi (Hu & Sadigh, 2023), and Minecraft (Wang et al., 2023a), or be deployed on robots for real-world interactions (Ahn et al., 2022; Driess et al., 2023). Recently, large multimodal models have garnered significant attention due to their ability to process various input modalities (text, images, videos, etc.). This has spurred increased research in embodied AI, where language-vision models are employed for decision-making and task planning in both simulated environments and the real physical world. While LLMs and Vision-Language Models (VLMs) can provide insightful suggestions for complex generation tasks, they often fail in solving simple decision-making tasks due to misalignment issues (Ahn et al., 2022).

To enhance decision-making capabilities, utilizing Chain-of-Thought (COT) reasoning has become a common approach. COT has been demonstrated to improve model performance in logical reasoning by facilitating the output of correct results through step-by-step reasoning. Mu et al. (2023) enhanced static planning capabilities by fine-tuning models on the EgoCOT dataset, integrating high-level task planning with low-level task control in a closed-loop manner, achieving promising performance in multiple specific tasks. In contrast, dynamic re-planning for decision-making has been shown to be more adaptive than static generation. Song et al. (2023b) introduced a few-shot planning method

leveraging in-context learning and a grounded re-planning mechanism to dynamically adjust high-level plans based on environmental observations.

Nevertheless, planning based solely on a model's generative capabilities is insufficient, especially in complex tasks, partially observable scenarios, and multi-task environments. Agent models must possess the ability for continual learning, continuously deriving insights from failures and aligning online within specific task environments to make more accurate decisions. Aligning through reinforcement learning (RL) is a common approach. RL learns agents' policies from scratch through trial and error in environments (Sutton, 2018), ensuring that LMM-based agents are well-aligned with their environments. A notable example is RL4VLM (Zhai et al., 2024), which combines Proximal Policy Optimization (PPO) with COT reasoning to fine-tune vision-language models for decision-making tasks. This integration allows the model to learn more effectively from task rewards through interaction, improving exploration, adaptability, and reasoning. RL4VLM proposes to mitigate the effect of the COT reasoning tokens by focusing the primary optimization target on the final action tokens. Most RL methods start with random policies, which are updated based on returns from the environment. This leads to poor sample efficiency, as initial policies perform poorly during the early stages of learning. One way to improve sample efficiency is to incorporate prior knowledge into the policy initialization and exploration during training (Kumar et al., 2022). LLMs are ideal sources of prior knowledge for RL agents, as they are trained on vast amounts of data from diverse corpora. Therefore, leveraging RL to align LLMs with embodied environments for decision-making tasks can simultaneously address the misalignment issues in LLMs and the sample efficiency challenges in RL.

Unlike RL4VLM, we believe that the COT process holds the key to optimization. Since the action is the outcome of the COT process and is closely related to it, we focus more on the consistency between the action and the COT. Our framework is based on Direct Preference Optimization (DPO). DPO has recently emerged as a prominent method due to its efficient alignment without the need for reward design, and it is widely used in the post-SFT stage of large models. To our knowledge, there is no precedent for its use in embodied agent tasks. Therefore, we consider introducing the DPO algorithm to efficiently learn strategies from sparse or no-reward interactions. Furthermore, we have made improvements to DPO with a focus on optimizing the COT process and ensuring consistency in model responses, thereby adapting it to our algorithmic framework for interactive alignment of VLMs.

In summary, our contributions can be summarized in the following four points:

1. We propose an algorithmic framework, **InteractiveCOT**, for online alignment of the COT process in embodied agents through interaction with the environment, supporting both PPO and DPO alignment schemes.

2. We have made adaptive adjustments to DPO, designing a data sampling and sample pair construction framework tailored to the interaction characteristics of embodied agents, thereby improving the sample utilization efficiency of the alignment algorithm.

3. We emphasize that aligning the COT is more important than aligning the final action in alignment tasks. Based on this, we have improved the DPO algorithm to enhance output consistency, alleviating the issue of output degradation during model training.

4. We validate our approach through experiments in the ALFWorld environment, demonstrating a **6%** increase in average success rates compared to baseline methods. Our results highlight the potential of integrating preference-based reinforcement learning techniques with COT processes to enhance the decision-making capabilities of vision-language models in embodied agents. This advancement highlights the importance of optimizing the thought process itself to achieve better performance and adaptability in complex, dynamic tasks.

## 2 RELATED WORK

**Embodied Agent with LLMs**  The successful integration of language as a semantically rich input for interactive decision-making highlights the crucial role of LLMs in facilitating interaction and decision-making processes (Abramson et al., 2020; Karamcheti et al., 2022; Li et al., 2022). LLMs are also applied in various environments to aid robot navigation (Parisi et al., 2022; Hong et al.,

2021; Majumdar et al., 2020) and manipulation (Jiang et al., 2022; Ren et al., 2023; Karamcheti et al., 2022). Recently, there have been a large number of methods that utilize LLMs to enhance agents' planning and reasoning capabilities in embodied agents. SayCan (Ahn et al., 2022) assesses the affordance of candidate actions by multiplying their probabilities under LLMs with a value function. (Zeng et al., 2022) combine the LLM with a visual-language model and a pretrained language-conditioned policy (Shridhar et al., 2022) to enable open vocabulary robotic tasks. (Huang et al., 2022a) demonstrate that LLMs can be employed for planning and executing simple household tasks. They ground LLM-generated actions by comparing their embeddings with a predefined list of acceptable actions. To incorporate environment feedback, Inner Monologue (Huang et al., 2022b) extends SayCan using a closed-loop principle. This principle is also applied in related works such as (Yao et al., 2023; Huang et al., 2022b; Kim et al., 2024; Singh et al., 2023; Liang et al., 2023; Shinn et al., 2023; Wang et al., 2023b) to continuously monitor agent behaviors and refine and adjust plans accordingly for tasks such as computer automation, Minecraft, etc. Furthermore, there are approaches that prompt LLMs to generate temporal-abstracted actions (Zheng et al., 2023). (Dasgupta et al., 2023) employ the LLM as a planner and success detector for an agent with their actor module necessitates pre-training with RL to enable the agent to follow natural language instructions. While these works demonstrate impressive results, they rely too heavily on the inherent capabilities of powerful LLMs, like GPT4 and PaLM (Chowdhery et al., 2023), which are difficult to apply to smaller LLMs with weaker reasoning abilities, like LLaMA-7B.

Concurrent to our work, GLAM (Carta et al., 2023) utilizes RL finetuning to achieve functional grounding of LLMs. However, they focus on simple primitive actions (turn left, turn right, go forward, etc.) evaluated in toy environments, BabyAI (Chevalier-Boisvert et al., 2018) with a much smaller encoder-decoder LLM, Flan-T5-780M. These primitive actions have a similar number of tokens and less meaningful semantics, resulting in underutilizing the capabilities of LLMs, and failing to observe the impact of prompt design and address the unbalance over action space, resulting in additional instability and poor robustness.

**Preference Learning** Preference learning has become a pivotal area in machine learning, aiming to develop predictive models that capture human preferences from observational data. Recent advances in deep learning and optimization algorithms have driven significant progress in this field, particularly in applications such as recommender systems, information retrieval, and personalized user interfaces.

Current preference learning methods can be categorized into pointwise, pairwise, and listwise approaches. Among these, Direct Preference Optimization (DPO) (Rafailov et al., 2024) has emerged as a novel and efficient paradigm, directly optimizing user preferences without intermediary ranking steps. DPO achieves more precise alignment with user preferences by constructing loss functions that directly reflect these preferences. Chen et al. (2024) introduces OPTune, an efficient method for online preference tuning in RLHF. By selectively regenerating low-reward responses and using a weighted DPO loss to focus on response pairs with larger reward gaps, OPTune improves training speed and model alignment while reducing computational costs Recent pioneering studies have further expanded DPO's applications and effectiveness. Step-DPO Lai et al. (2024) stands out as a significant advancement over Direct Preference Optimization (DPO) for tasks requiring long-chain reasoning, such as mathematical problem-solving. Unlike DPO, Step-DPO optimizes individual reasoning steps. By focusing on pinpointing the first erroneous step in a sequence and optimizing for more fine-grained accuracy, Step-DPO improves both factuality and reasoning in large language models. Pal et al. (2024), in their DPO-Positive study, advanced practical applications of DPO by focusing on positive direct preference optimization in sentiment-aware recommendations. The DPO-Positive method not only enhances user satisfaction but also incorporates sentiment information into the recommendation process, resulting in more accurate and user-aligned outcomes.

## 3 METHODS

### 3.1 ONLINE TRAINING OF REPLANNING FRAMEWORK

In previous work (Ahn et al., 2022; Song et al., 2023a), long-term planning using large language models (LLM) or large multimodal models (LMM) has typically been approached as static planning, the transition and planning between the initial and final states of a task is accomplished

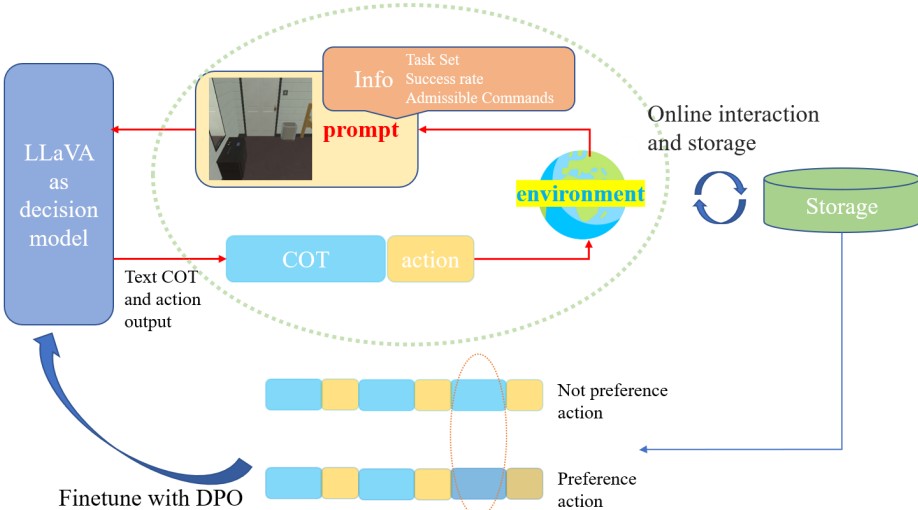

Figure 1: Main framework of our method. In our approach, we sample two different trajectories for the same stepx and assess their preference based on task completion rates. This allows us to determine the preference level of different actions in the current state. We then fine-tune the vision-language model (VLM) using preference methods such as Direct Preference Optimization (DPO).

through a single planning instance. For once planning instance, planner output the planning results $\{planner : s_0, a_1, a_2, \cdots, a_n, s_{goal}\}$, where $s$ denote states and $a$ for actions. This static planning often considers the completeness of adjacent decisions and planning costs. However, it frequently lacks the capability to timely correct erroneous plans. Consequently, the feedback provided by the environment following each planning action may not be utilized in a timely manner to adjust subsequent actions. The primary distinction between re-planning and static planning processes lies in the ability to make timely adjustments based on environmental feedback. Re-planning captures factors that change dynamically within the environment, providing different responses based on various states on each decision-making steps and generating new execution plans. Compared to static planning, this approach offers greater adaptability and robustness. Additionally, dynamic planning involves deeper interaction with the environment.

Our algorithm is designed based on the re-planning framework, which can be seen in Figure 1. Specifically, in each natural step of interaction between the agent and the environment, the planning result of next several time-steps is regenerated according to the current observation. We incorporate camera images and environmental feedback into the design of prompts, providing feedback to the agent at each step, requiring it to give subsequent plans step-by-step based on observations. Under the re-planning framework, a well-fine-tuned model base can already perform quite well. However, there are still some complex situations that the agent cannot handle effectively, such as navigation tasks where target objects are not observable and complex tasks with numerous steps. We ponder *whether the agent can learn and improve its planning abilities through interaction with the environment*. Reinforcement learning algorithms are a good choice, but they require a precisely designed reward function and must also consider potential reward hacking phenomena. Based on this, our framework aligns through the construction of preference sample pairs. During the interaction process, we sample and use the success rate of trajectories as a preference for alignment. When applying DPO to fintune VLM, the loss function is

$$L(\theta) = -\mathbb{E}_\zeta \left[ \log \sigma \left( \beta \log \frac{\pi_\theta(a_{win}^t | \tau_1^{t-1})}{\pi_{ref}(a_{win}^t | \tau_1^{t-1})} - \beta \log \frac{\pi_\theta(a_{lose}^t | \tau_2^{t-1})}{\pi_{ref}(a_{lose}^t | \tau_2^{t-1})} \right) \right] \tag{1}$$

where $\zeta$ is a tuple of $(o_t, s_{t-1}, a_{win}, a_{lose})$, $s_{1 \sim t-1}$ is the state-action decision trajectory from time 0 to time $t-1$, $a_{win}$ and $a_{lose}$ represent the preference and not preference decision-making actions at current timestep $t$. $\pi_\theta$ and $\pi_{ref}$ denote the policy generated by VLM, $\pi_{ref}$ refers to the policy generated by the unrefined output of VLM. This approach enables a more granular alignment of

preferences, focusing on fine-tuning for each individual decision rather than aligning preferences for the entire trajectory. Consequently, while introducing step-wise preference information, we improve upon previous methods that used the PPO algorithm for fine-tuning VLMs by adopting DPO for preference alignment. This allows us to align preferences for different decisions at each segment of the trajectory.

## 3.2 TRAINING COT WITH DPO

In completing each task, the input prompt for the VLM includes observations and action trajectories. Through the design of input prompts and instruction-following mechanisms, VLM can generate decision actions based on the current state, produce feasible actions, and provide a textual description of the current observation along with the formatted output. This structured approach enables the model to maintain context and make informed decisions effectively. The response output by the VLM includes a chain-of-thought reasoning process for the current action, immediately followed by the keyword "action", which indicates the model's current decision action. This structured format allows for clarity in the decision-making process, ensuring that the reasoning is explicitly linked to the chosen action.

### 3.2.1 THE METHOD OF CONSTRUCTING SAMPLE PAIRS

Compared to classic MLP-based policy networks, a advantage of VLM policies is that they can output neutral language, thus leverage COT reasoning for efficient exploration by performing intermediate reasoning steps that lead to the final decision. However, training a VLM policy $\pi_\theta$ with RL presents additional challenges. First, due to the sparse rewards obtained from the online interactions between VLM and the environment, many state transition processes receive a reward feedback scalar value of $0$. In the case of state transition samples with a reward value of $0$, employing the PPO (Proximal Policy Optimization) architecture for fine-tuning the VLM makes it challenging for the model to learn effective strategies for interacting with the environment. Consequently, the sample efficiency of fine-tuning the VLM using these state transition samples is relatively low. In some studies, researchers often design reward functions manually to mitigate the issue of sparse rewards. On the other hand, preference-based methods can construct preference pairs using different reward values $\left\{ \tau_{win}^t = \left\{ a_t^1, r_t^1, \tau_1^{t-1} \right\}, \tau_{lose}^t = \left\{ a_t^2, r_t^2, \tau_1^{t-1} \right\} \right\}$, whereby trajectories with higher reward values can be treated as preferred trajectories, for example, $\tau_{win}$ has a higher reward $r_t^1$ and $\tau_{lose}$ has a lower reward $r_t^2$. This approach allows for a more nuanced representation of preferences, facilitating the learning process in environments characterized by sparse feedback. By employing this method, preference-based approaches can effectively leverage state transition samples with lower reward values, thereby enhancing sample efficiency. This strategy allows the model to learn from a broader range of experiences, improving its ability to identify and optimize preferred trajectories within the environment.

### 3.2.2 THINKING IS MORE IMPORTANT THAN DECISION-MAKING

It is worth noting that by outputting the text of the chain of thought, we enable the VLM to produce reasonable actions through autonomous reasoning. However, the reasoning ability of the VLM stems from its training on massive datasets. Jointly fine-tuning the COT text and the text actions output by the VLM is proven to be a better method adapting the VLM to embodied scenarios. This method often focuses on optimizing action output, and when designing loss functions, it tends to minimize the impact of the COT process or only consider the final action decision. The fine-tuning process breaks the coherence of the language output formed during pre-training, leading to model collapse. To solve this problem, we used two methods to constrain the fine-tuning process. First, we constrain the distributional distance between the fine-tuned output text and the unfine-tuned reference model output text, thus ensuring that the model does not deviate too much from the logicality of the original model language output due to fine-tuning. Similar to the derivation process of the DPO model, we set

$$Q(s, a) = \beta \log \frac{\pi_\theta(a|s)}{\pi_{ref}(a|s)} \tag{2}$$

Then, we can fine-tune the output strategy of VLM by optimizing the $Q$ value, while limiting the output distance between the fine-tuned model and the reference model without fine-tuning by adding

a regularization term of KL divergence to the optimization objective, which is as follows

$$\max_{\pi_\theta} \mathbb{E}_{s \sim D, a \sim \pi_\theta(a|s)}[Q(s,a)] - \beta D_{\text{KL}}[\pi_\theta \parallel \pi_{ref}] \tag{3}$$

Based on this optimization objective, combined with some mathematical derivations of Yang et al. (2024), we can derive the following step-wise optimization formula:

$$L = -\mathbb{E}_\zeta \log \sigma \left( \beta \log \frac{p(a_1^t | \mathcal{T}_1^t) \pi_\theta(\mathcal{T}_1^t | \tau_1^{t-1})}{\pi_{ref}(a_1^t, \mathcal{T}_1^t | \tau_1^{t-1})} - \beta \log \frac{p(a_2^t | \mathcal{T}_2^t) \pi_\theta(\mathcal{T}_2^t | \tau_2^{t-1})}{\pi_{ref}(a_2^t, \mathcal{T}_2^t | \tau_2^{t-1})} \right) \tag{4}$$

$$\nabla_\theta L = -\beta \mathbb{E}_\zeta [\Lambda[\nabla_\theta \log \pi_\theta(\mathcal{T}_1^t | \tau_1^{t-1}) - \nabla_\theta \log \pi_\theta(\mathcal{T}_2^t | \tau_2^{t-1})]] \tag{5}$$

The detailed derivation can be found in Appendix A. In the formula, $\mathcal{T}_i^t$ is the output text of the thinking chain at step $t$, and $a$ is the output action after the thinking chain. However, by calculating the gradient of 4, we can see that the gradient term in Equation 5 directly eliminates the influence of action probabilities, where $\Lambda = \sigma(\hat{Q}_\theta(a_1^t, \mathcal{T}_1^t, \tau_1^{t-1}) - \hat{Q}_\theta(a_2^t, \mathcal{T}_2^t, \tau_2^{t-1}))$. Therefore, in practice, we adopt the following action probability weighting (APW) form:

$$\tilde{L} = -\mathbb{E}_\zeta \log \sigma \left( \beta p(a_1^t | \mathcal{T}_1^t) \log \frac{\pi_\theta(\mathcal{T}_1^t | \tau_1^{t-1})}{\pi_{ref}(a_1^t, \mathcal{T}_1^t | \tau_1^{t-1})} - \beta p(a_2^t | \mathcal{T}_2^t) \log \frac{\pi_\theta(\mathcal{T}_2^t | \tau_2^{t-1})}{\pi_{ref}(a_2^t, \mathcal{T}_2^t | \tau_2^{t-1})} \right) \tag{6}$$

We will later analyze the errors of both and demonstrate that our approach is feasible. Intuitively, the gradient term of the "action" probability adds weight to the "thoughts" probability. Actions with higher output probabilities after COT indicate a better alignment with the thoughts process, while lower probabilities suggest greater randomness in action generation. The weighting term can reduce the generation of highly random positive samples and encourage the generation of deterministic positive samples.

### 3.2.3 MAKE DECISIONS THAT ARE MORE CONSISTENT WITH REASONING

In the second method, we consider adding a regularization term to the final action text output. Fine-tuning the reasoning chain may alter the model's language output conventions, potentially leading to model collapse. Aligning the action text output with the reference standard model ensures that the model adheres to the prompt's formatting requirements, thus generating valid actions. This approach helps maintain the integrity of the output while allowing for effective task execution, the effects of this regularization can be observed in Figure 2.

To further ensure the consistency between the COT process of generating text and the final action, we consider adding a stronger constraint to the above formula. We believe that the pre-trained model has already been well-optimized for modeling the process from thoughts to actions. Therefore, in the subsequent interaction phase, we will maintain alignment with the pre-training results in the dimension of generating actions based on thoughts. We will add a mean square error (MSE) regularization term of action policy consistency constraint (APC), specifically:

$$L_{InteractiveCOT} = \tilde{L} + \kappa \text{MSE}(\pi_\theta(a_1^t | \mathcal{T}_1^t), \pi_{ref}(a_1^t | \mathcal{T}_1^t)) \tag{7}$$

Where $\kappa$ is a hyper-parameter that regulates the strength of constraints. Note that this is different from the KL divergence used in the DPO derivation process with the reference model; Here, the focus is more on aligning the process of deriving actions from thoughts. In the comparison experiment of action consistency constraint, we extracted an output sample at the training step of 2000, both with and without the action probability consistency constraint, as shown in Figure 2. The sample clearly demonstrates the difference between the COT process and the final decision action output. In the sample with the action consistency constraint, the model's COT process provides a clear analysis and identifies the valid action to be taken next. In contrast, in the sample without the consistency constraint, although the agent provides reasonable thoughts and analysis, it ultimately outputs an irrelevant action in the final decision, which does not fall within the scope of valid actions.

Finally, we provide a simple illustration of Equation 6 to demonstrate that our approximation is reasonable. We assume that the pre-trained model has achieved good alignment, so $p(a|\mathcal{T})$ will be close to 1. We have:

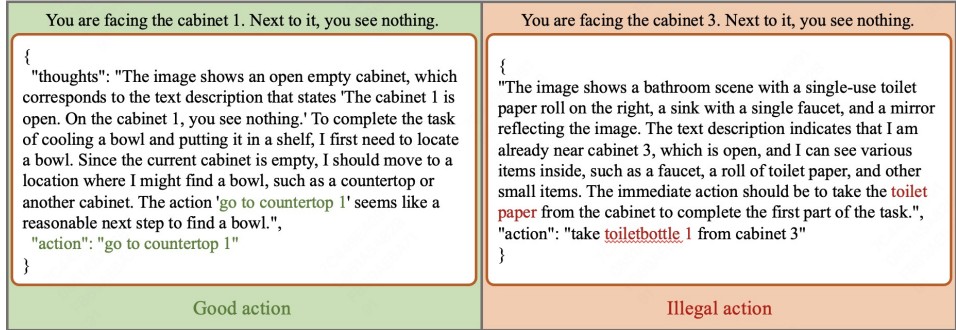

Figure 2: Use or not use action policy consistency constraint cases. When employing the use action policy consistency constraint strategy, the output actions are more likely to be valid actions. This approach helps to ensure that the generated actions align with the established policy, thereby enhancing the reliability and appropriateness of the actions in the context of the task being performed.

$$\Delta(p(a_i^t|\mathcal{T}_i^t)) = \log \frac{p(a_i^t|\mathcal{T}_i^t)\pi_\theta(\mathcal{T}_i^t|\tau_i^{t-1})}{\pi_{ref}(a_i^t, \mathcal{T}_i^t|\tau_i^{t-1})} - p(a_i^t|\mathcal{T}_i^t)\log\frac{\pi_\theta(\mathcal{T}_i^t|\tau_i^{t-1})}{\pi_{ref}(a_i^t, \mathcal{T}_i^t|\tau_i^{t-1})} \tag{8}$$

$$= \log p(a_i^t|\mathcal{T}_i^t) + (1 - p(a_i^t|\mathcal{T}_i^t))\log\frac{\pi_\theta(\mathcal{T}_i^t|\tau_i^{t-1})}{\pi_{ref}(a_i^t, \mathcal{T}_i^t|\tau_i^{t-1})} \tag{9}$$

This variable will approach zero as $p(a|\mathcal{T})$ approaches 1. In practice, we have calculated the approximate distribution of action probabilities and demonstrated that our assumption is well-founded, which can be shown in Figure 6b.

## 4 EXPERIMENTS

In this part we perform experiments to validate three key questions:

- How does our framework enhance the decision-making capabilities of VLM?
- Can the regularization term effectively constrain the action distribution to prevent deviation from the original policy?
- Does action-weighting mitigate the issue of degradation?

we conducted experiments in the ALFWorld environment and recorded improvements in the visual semantic reasoning capabilities of the Vision-Language Model. ALFWorld encompasses six types of household planning tasks: *Pick & Place , Pick Two & Place, Clean & Place, Cool & Place, Heat & Place,* and *Examine in Light*. For convenience, we will refer to them as **Pick**, **Pick2**, **Clean**, **Cool**, **Heat** and **Look** hereafter. During the experiments, the agent captures a visual observation through egocentric view in the current state and a textual instruction describing the task to be completed. The agent must plan and navigate based on the visual information to accomplish the specified tasks. We instantiate our method on top of the *llava-v1.6-mistral-7b* model, and build the agent based on this model. During interactive, we package the observation picture into a special prompt to get LLaVA's answer.

**Prompt**  Our COT prompt consists of the following parts: First, we clarify the task requirements. The tasks in ALFWorld are semantically rearranged. For example, both "examine the pillow with the desklam" and "look at the pillow under the desklam" indicate that the agent needs to find the pillow, pick it up, then locate and navigate to the desklamp. Secondly, we specify the range of valid actions. Each state in ALFWorld environment is accompanied by different valid action transitions. For instance, if the action is to pick up pillow 1, the prerequisite is that the agent must be close enough to reach the pillow. If the action is to put down an object, it must have previously executed the pick-up action. Therefore, one of the criteria for evaluating the agent's capability in the experiment

is whether the actions it outputs are valid. Finally, we specify the output format of the agent's LLaVA model, which must strictly follow the JSON format containing "thoughts" and "action". The action must be derived from the thoughts and should not produce irrelevant actions. Our prompt design is shown in Figure 3:

> Your are an expert in the ALFRED Embodied Environment.
> Your task is to ∗ *task name* ∗. You are also given the following text description of the current scene: ∗ *obs* ∗}.
> Your admissible actions of the current situation are: [∗ *reformatted admissible actions* ∗].
> Your response should be a valid Json file in the following format:
> "thoughts": "{first describe what do you see in the image using the text description, then carefully think about which action to complete the task. },
> "reflections": "{reflect on your historical trajectory and carefully think about which action to complete the task.}",
> "action": "{an admissible action}"
> your actions should be based solely on the analysis provided by your thoughts!
> your output need to be in 60 words!

Figure 3: Prompt used in ALFWorld tasks. The prompt provides the embodied agent with several key components: the task to be completed, the current egocentric observations, the feasible actions available, the output format for the thought process in the reasoning chain, the format for action text output, and constraints on the length of the output text. This structured approach helps ensure that the VLM can generate coherent and contextually relevant responses, facilitating effective decision-making and task execution.

**Implementation**  Our implementation consists of two parts. First, we conduct one epoch of model SFT (supervised fine-tuning) on the open-source dataset LEVI-Project/sft-data (Zhai et al., 2024) to ensure the model's ability of formatted output . The LEVI-Project/sft-data dataset is an expert trajectory dataset sampled by a GPT-4-based agent, containing 45k different state samples, each adhering to the JSON format of COT outputs. After SFT, we employ the model to interact with the environment, optimizing its COT capabilities during these interactions and monitoring performance in real-time during training.

### 4.1 How much better we are at making decisions

The aim of experiments in this section is to validate the performance of the InteractiveCOT method. To evaluate whether the algorithm can consistently generate decisions through the COT process, we use the success rate of task execution as a reference and select PPO from the RL4VLM (Zhai et al., 2024) framework as the baseline. ALFWorld does not provide a reward function during interactions; it only indicates whether the current task is successfully executed and returns the task's progress. For instance, if a task requires checking an object under a table lamp, finding and picking up the object results in a 50% progress update. Given that such progress updates are sparse in a larger action space, we construct preference criteria for preference learning. The preference score for each trajectory is calculated using Equation 10:

$$P = 50 * success\ rate - \mathbb{1}_{\{invalid\}} \tag{10}$$

$$\mathbb{1}_{\{invalid\}} = \begin{cases} 1 & \text{if } action \text{ not in } admissible\ action \\ 0 & \text{otherwise} \end{cases}$$

where $\mathbb{1}_{\{invalid\}}$ represents our stronger rejection of illegal actions given the same success rate. During the exploration phase, the agent collects trajectory data and constructs sample pairs based on the six task types mentioned above. Higher preference scores indicate greater sample preference. In practice, considering the achievement of long-term goals, we calculate preference scores using a method similar to discount factor weighting in reinforcement learning returns. Due to the high randomness of ALFWorld, we set up experimental environments with different seeds and calculated mean and variance of each results.

We use Equation 7 for the model weight update with $\kappa = 0.1$, measure the agent's performance by the average success rate of each task. The final results are shown in Figure 4.

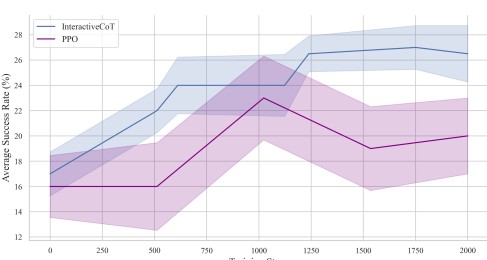 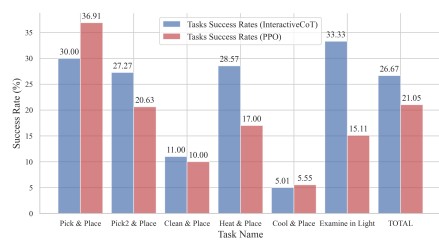

(a) Average success rate during training (all 6 tasks)    (b) Average success rate of each task at 2000 steps

Figure 4: We demonstrate that fine-tuning the vision-language model (VLM) using Interactive Chain of Thought (InteractiveCOT) and Proximal Policy Optimization (PPO) results in varying task completion rates and average task completion rates in ALFworld. Our findings indicate that, for the majority of tasks, fine-tuning the model using preference methods yields better results than using reinforcement learning approaches. Additionally, we observe that through online interaction with the environment, the preference method achieves the same average task completion rate with fewer interaction steps, indicating higher sample efficiency and more minimal model degradation.

ALFWorld gives task randomly so we calculate the overall success rate as the weighted average of success rates under all tasks. InteractiveCOT shows an improvement in the overall success rate, indicating that our algorithm can learn more efficiently from interactions. In our experiments, we used approximations such as $\log(\pi(a|\mathcal{T},\tau)\pi(\mathcal{T}|\tau)) \approx \pi(a|\mathcal{T},\tau)\log(\pi(\mathcal{T}|\tau))$ when $\pi(a|\mathcal{T},\tau) \to 1$, We calculated the occurrence probability distribution of action tokens in the experiments to demonstrate that our approximations are reasonable.

## 4.2 WHAT ROLE DOES ACTION POLICY CONSISTENCY CONSTRAINT PLAY?

We pointed out that during training, to enhance stability, we introduced the regularization of action token probabilities between finetune model and reference model. This section will explore the impact of regularization on the results and investigate its role. We designed ablation experiments, where we conducted trials with different regularization weight values $\kappa$ under the same parameter settings, and recorded the average success rate of the agent during training. In this experiment, we use Equation 1 with the regular term as the loss function, with other conditions the same as in Section 4.1.

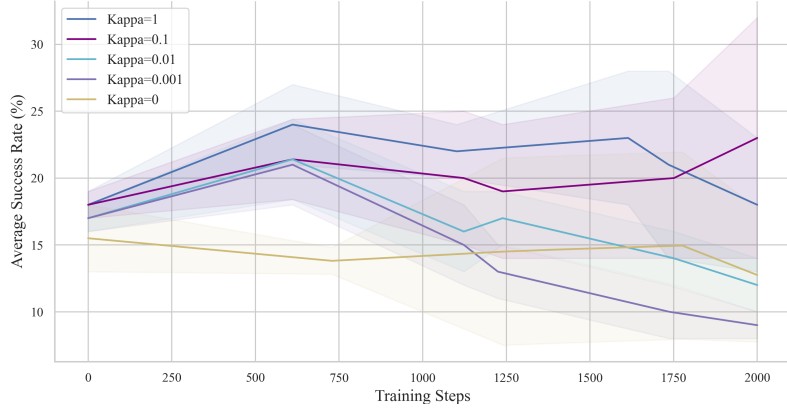

Figure 5: Parameter study of APC

The results in Figure 5 show that different values of $\kappa$ significantly impact the success rate. As the parameter increases, the action policy consistency constraint strengthens, leading to improved model performance. This validates the importance of regularization. However, when $\kappa$ is set to 1, the algorithm's performance declines, indicating that $\kappa$ should neither be too large nor too small, with

a value around 0.1 yielding near-optimal performance. Given the importance of the $\kappa$ parameter, its optimal value may vary across different environments or tasks. Due to space constraints, we do not explore this further in this paper.

### 4.3 MORE CERTAIN, MORE STABLE

In Section 3.2.2, we mentioned that the gradient weighted by action probability would prefer more certain successful strategies, which indirectly achieves the unification of thoughts and actions—the larger the conditional probability of an action, the more closely it is linked to the content of the COT. To verify this idea, we conducted following ablation experiments. We used loss functions with and without action probability weighting, Equation 6 and Equation 4, keeping all other settings identical to the main experiment. Figure 6a shows the comparison between the two sets of experiments, and Figure 6b presents the probability distribution of all actions in the first 2000 training steps. It is evident that under the APW condition, the probability distribution of action tokens is mostly concentrated around 1, indicating more certain and robust decision-making, which also leads to a higher success rate. In contrast, the results without weighting show a more dispersed distribution of action probabilities, with some probabilities falling below 0.8, which is not conducive to the convergence of the algorithm.

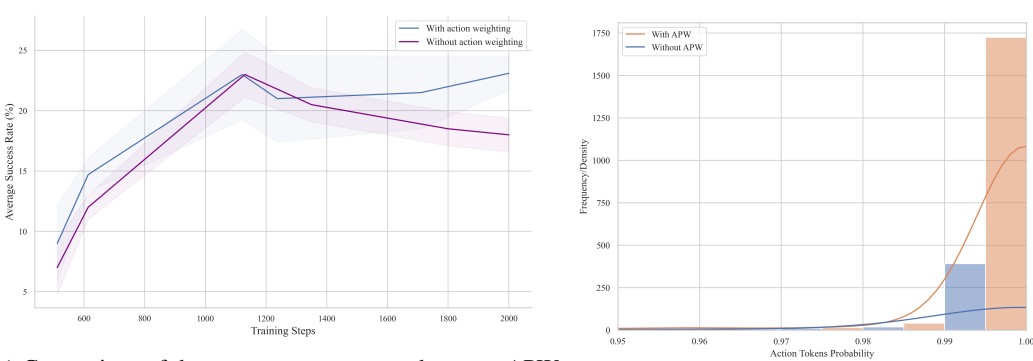

(a) Comparison of the average success rates between APW and non-APW methods

(b) Action tokens probability distribution

Figure 6: Validate the impact of APW in interactions.

## 5 CONCLUSIONS, LIMITATIONS AND FUTURE DIRECTIONS

This work introduces an algorithmic framework, InteractiveCOT, for online interactive fine-tuning of multimodal models during the COT process, supporting both PPO and DPO algorithms. Based on LLaVA-7B, we execute household tasks in embodied scenarios through dynamic replan, achieving better decision-making by aligning COT capabilities. We emphasize the core importance of COT, moving away from previous approaches that primarily focused on training actions. Instead, we maintain the consistency between COT and actions through APW and APC. Empirical results demonstrate that InteractiveCOT outperforms reinforcement learning algorithms in average performance within ALFWorld. Ablation studies further confirm the critical role of APW and APC in the algorithm's convergence effectiveness.

One limitation of this study is the lack of validation across a broader range of environments and tasks, which will be addressed in future work. We aim to further optimize the generalization performance. Another limitation is the consideration of non-Markovian processes. Since the pre-training datasets in the SFT phase are all Markovian, our interaction experiments were conducted under the same conditions. Non-Markovian processes are more common in complex decision-making tasks, and effectively handling historical information is a crucial capability for agents. In future work, we will first deploy our algorithm framework in more simulated environments and datasets to enrich the experimental results. Additionally, we will consider modeling non-Markovian processes, focusing on the agent's performance with long historical information.

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

## A DERIVATION OF FORMULAS

We provide a simple derivation of Equation 4. During the RL phase with reward model, the object of training is to maximize returns. Following prior works the optimization is formulated as:

$$\max_{\pi_\theta} \mathbb{E}_{s \sim D, a \sim \pi_\theta(a|s)}[Q(s,a)] - \beta D_{KL}[\pi_\theta \parallel \pi_{ref}] \tag{11}$$

which can be rewritten as:

$$\max_{\pi_\theta} \mathbb{E}_{s \sim D, a \sim \pi_\theta(a|s)}[Q(s,a)] - \beta D_{KL}[\pi_\theta \parallel \pi_{ref}]$$

$$= \max_{\pi_\theta} \mathbb{E}_{s \sim D, a \sim \pi_\theta(a|s)}\left[Q(s,a) - \beta \log \frac{\pi(a|s)}{\pi_{ref}(a|s)}\right]$$

$$= \min_{\pi_\theta} \mathbb{E}_{s \sim D, a \sim \pi_\theta(a|s)}\left[\log \frac{\pi(a|s)}{\pi_{ref}(a|s)} - \frac{1}{\beta}Q(s,a)\right]$$

$$= \min_{\pi_\theta} \mathbb{E}_{s \sim D, a \sim \pi_\theta(a|s)}\left[\log \frac{\pi(a|s)}{\pi_{ref}(a|s)\exp\left(\frac{1}{\beta}Q(s,a)\right)}\right]$$

$$= \min_{\pi_\theta} \mathbb{E}_{s \sim D}[D_{KL}[\pi(a|s) \parallel \tilde{\pi}(a|s)]]$$

where $\tilde{\pi}(a|s) = \pi_{ref}(a|s)\exp\left(\frac{1}{\beta}Q(s,a)\right)$. KL-divergence is minimized at zero if and only if the two distributions are identical. Therefore, in the case of the optimal solution we get:

$$\pi(a|s) = \tilde{\pi}(a|s) = \pi_{ref}(a|s)\exp\left(\frac{1}{\beta}Q(s,a)\right)$$

A simple transformation yields:

$$Q(s,a) = \beta \log \frac{\pi(a|s)}{\pi_{ref}(a|s)} \tag{12}$$

We can know from Yang et al. (2024) that the Q-value form of Bradley-Terry preference distribution can be expressed as:

$$p(\tau_1 > \tau_2 | a_i^t, s_i^t, a_i^{t-1} ..., s_i^0)_{i \in \{1,2\}} = \frac{\exp(Q(s_1^t, a_1^t))}{\sum_{i \in \{1,2\}} \exp\left(Q(s_i^t, a_i^t)\right)} \tag{13}$$

Combining Eq. 12 and Eq. 13, replacing $s_i^t$ with $\tau_i^{t-1}$ and $a_i^t$ with $(a_i^t, \mathcal{T}_i^t)$, we derive the following loss function:

$$L = -\mathbb{E}_\zeta \log \sigma \left(\beta \log \frac{\pi_\theta(a_1^t, \mathcal{T}_1^t | \tau_1^{t-1})}{\pi_{ref}(a_1^t, \mathcal{T}_1^t | \tau_1^{t-1})} - \beta \log \frac{\pi_\theta(a_2^t, \mathcal{T}_2^t | \tau_2^{t-1})}{\pi_{ref}(a_2^t, \mathcal{T}_2^t | \tau_2^{t-1})}\right) \tag{14}$$

which is similar to Eq. 4

