# OpenReview forum: "InteractiveCOT: Aligning Dynamic Chain-of-Thought Planning for Embodied Decision-Making"
_ICLR.cc/2025/Conference — Submitted to ICLR 2025_

### Official Review · Reviewer_fL8F · 2024-10-29

**Soundness:** 2
**Presentation:** 2
**Contribution:** 2
**Rating:** 3
**Confidence:** 3

**Summary:**

This paper introduces InteractiveCOT, a method for fine-tuning a vision-language model (LLaVA-7B) for embodied AI tasks. The authors aim to enhance RL4VLM by optimizing the chain-of-thought (CoT) process rather than focusing solely on final action decisions. Using the Direct Preference Optimization (DPO) algorithm, they fine-tune the vision-language model with online-collected episodes. The proposed method is evaluated in the ALFWorld environment, demonstrating improvements over the PPO baseline.

**Strengths:**

- Utilizing foundation models as a starting point for embodied agents is an interesting approach, as it can enhance sample efficiency and reduce redundant exploration by leveraging the prior knowledge embedded in these models.
- InteractiveCOT achieves a 6% improvement in the final success rate by focusing more on optimizing the CoT process during VLM fine-tuning.

**Weaknesses:**

- The paper's presentation is difficult to follow, with numerous typos and inconsistencies in notation. For example:
    - Line 84, "In summery" -> "In summary".
    - In Figure 1, "LLaVA as dicision model" -> "LLaVA as decision model."
    - Line 215, "donate" should be "denote"; additionally, $\pi_{ref}$ is duplicated.
    - The definitions of subscripts and superscripts for action (i.e., $a_t^1$ and $a_t^2$) in line 245 and in Equations (4), (6), (7), (8), and (9) are inconsistent.
    - Line 213 references the tuple with $o_t$, but it is unclear where $s_{1 \sim t-1}$ originates.

- The authors should include a background section to introduce the basic RL framework, including elements of the MDP, trajectories, and policy, to clarify the RL context being considered. Without this, it is difficult to follow the subsequent sections. Additionally, a brief overview of the original DPO algorithm should be provided so that modifications proposed in the methods section are clearly distinguishable.

- In Section 3.1, the authors state that the VLM is used as a planner; however, it is unclear how the plan is generated. It appears that the VLM functions directly as a policy, outputting final actions to step into the environment, as illustrated in Figure 1. Thus, it may be misleading to frame the proposed method as a "re-planning framework" (line 197). Can author clarify this point?

- What type of action space does the paper consider? Is it continuous or discrete? If it is discrete, how is the MSE calculated in Eq. (7)?

- In line 201, what does "well-fine-tuned model" refer to? Is this the VLM fine-tuned by the proposed method?

- Throughout the paper, what does $\tau_t^{t-1}$ represent?

**Questions:**

See the weaknesses.

---

> ### Author Response · Authors · 2024-11-22
>
> Thank you very much for your thorough review and valuable suggestions. In response to the issues you raised, we provide the following clarifications and revisions:
>
> ---
>
> **Presentation Clarity of the Paper:**
> Regarding your concern that the paper's presentation is difficult to follow, we assure you that in the revised version, we will meticulously correct all spelling and grammatical errors. Specifically, concerning the subscript notation in the equations, in Equation (9), the subscript $i$ in $a_i$ denotes either a preferred or non-preferred action (i.e. $i =1$ or $2$). We will include additional explanations to clarify this notation.
>
> ---
>
> **Background Introduction on RL and DPO Algorithms:**
> You pointed out the need for a more comprehensive background on Reinforcement Learning (RL) and Direct Policy Optimization (DPO) algorithms. Due to space constraints, our initial submission focused more on the methodology and experimental sections. In the revised manuscript, we will incorporate a succinct yet thorough introduction to the relevant background, ensuring that readers gain a clear understanding of RL and DPO and their roles in our study.
>
> ---
>
> **Use of VLM as a Planner:**
> Regarding the use of Vision-Language Models (VLM) as planners, we conceptualize our VLM agent as a single-step planner. **Unlike static planning approaches, our VLM generates new analyses and plans at each step.** For every output, the VLM first analyzes the instructions and the current context to formulate a plan for the subsequent actions—this constitutes the "thoughts." Based on these thoughts, the VLM produces the necessary text actions to execute. In theory, our approach can support multi-step planning; however, to maintain consistency with baseline settings, we will explore this aspect in future work.
>
> ---
>
> **Types of Actions and Calculation of Mean Squared Error (MSE):**
> Our actions consist of discrete text actions formatted and output by the VLM. The MSE calculation targets **the probability distribution of action tokens** between the reference model and the updated model. Our objective is to ensure that the probability of generating actions in the new model remains consistent with the original model. This constraint of consistency ensures that during interactive training, **the logical correlation from thoughts to actions is preserved**, preventing the loss of this critical relationship.
>
> ---
>
> **Explanation of *Well-Fine-Tuned Model* in Line 201:**
> In response to your query about the term "well-fine-tuned model" mentioned on line 201, we provide further clarification. In Section 4 of the Implementation part of the paper, we detail our implementation specifics. Here, the "well-fine-tuned model" refers to a model that has undergone **Supervised Fine-Tuning (SFT) on a multimodal decision-making dataset**. Our method leverages this SFT model as the foundation for deploying reinforcement learning algorithms, thereby enhancing its generalization and adaptability capabilities.
>
> ---
>
> Once again, we sincerely thank you for your insightful suggestions. We are committed to addressing all the mentioned issues by incorporating the necessary additions and revisions to enhance the clarity, comprehensiveness, and overall quality of our paper.

---

> > ### Comment · Reviewer_fL8F · 2024-12-01
> >
> > Dear authors, after reading your rebuttal and other reviews, I believe the paper requires significant improvement in presentation clarity and structure. Additionally, including comparisons in domains beyond ALFWorld is essential to demonstrate the generalizability of the proposed method. Furthermore, since the action space is discrete, I do not believe using MSE is appropriate to keep two categorical distributions close. Hence, I will maintain my score.

---

### Official Review · Reviewer_r5ga · 2024-11-04

**Soundness:** 1
**Presentation:** 2
**Contribution:** 1
**Rating:** 3
**Confidence:** 3

**Summary:**

The paper proposes a preference based fine-tuning method for a VLM on sequential decision making tasks. They look at Alfworld tasks and show a 6% improvement over the baseline RL4VLM (which looks at RL based finetuning of VLMs). They add auxiliary losses (APW and APC) to encourage COT - action consistency in the output form the VLM.

**Strengths:**

How to benefit from and optimise chain of thoughts in LLM-derived models for decision-making is an interesting question which the paper considers.

**Weaknesses:**

The paper is very poorly written:
1) Many implementation details are left to the imagination, there isn’t a code implementation provided either.
2) The supplementary material only contains a single page derivation and no experimental details.
3) Lots of typos and confusing wording throughout the paper
4) Lines 214, 215, what is meant by “πref refers to the policy generated by the unrefined output of VLM? Donate → Denote
5) πθ and πref donate the policy generated by VLM - not clear enough
6) Line186  - typos? - For once(?) planning instance, planner output the …
7) Fig 1 : dicision -> decision
8) In many places there is mention of a “Preference action” - I suppose the authors mean to say “preferred action”
9) Fig 1 is not very clear - what is the circle trying to outline in the “preference action” and “not preference action”?
10) The subscripts in the equations are not defined anywhere
11) Line 222 typo in the section heading
12) The contributions claim interactiveCOT supports both PPO and DPO- where is this validated? Isn’t the paper only presenting DPO results?
13) Are their pairs action level or trajectory level
14) Point 2 in contributions: “tailored to the interaction characteristics of embodied agents, “ - what does this mean?

Lack of thoroughness in experiments:
1) The paper doesn’t mention what exactly is the reward function used by the PPO baseline? The paper mentions using partial task progress signal (which is much denser than a 0/1 success signal indicating whether the episode was successful) to construct preference labels - did the baseline (PPO) also benefit from this extra information? If not, why? It should be possible to convert the partial progress to rescaled rewards.
2) Fig 6a both variants seem the same in terms of the max performance resached yet the text describes one variant (w action weighting) as better than the other
3) No information about the hyperparameters is provided, no information on how many seeds were used, fig 4b does not even have error bars - was only one seed used per task??
4) How was the checkpoint used to report the final score chosen? Fig 4a has a larger gap between the two methods at 2k steps (6% as claimed by the authors), but the gap would be smaller (by around 2%) if you used the ckpt for PPO at 1k training steps - which would reduce the improvement to 4%.
5) How exactly were the trajectories collected?
6) No other baselines are studied apart from RL4VLM:  the RL4VLM paper cited studies a CNN+RL baseline, as well as a prompting-only LLM baseline. It would also make sense to compare to the baseline where the authors train a VLM with DPO directly for action selection (as done for the VLM-RL case in [1]).
7) What is the reason/motivation for using the particular preference score definition in eqn 10? Why 50 * progress  specifically?
8) 454 line - unclear, and where is this demonstrated?
9) Line 426 - extremely vague description of scores computation (“In practice, considering the achievement of long-term goals, we calculate preference scores using a method similar to discount factor weighting in reinforcement learning returns”)

[1] Large Language Models as Generalizable Policies for Embodied Tasks

**Questions:**

I have listed the questions in the bullet points of the weaknesses section.

---

> ### Author Response · Authors · 2024-11-22
>
> To begin with, we thank the reviewer for the carefully reviewing and insightful advice. Through your suggestions, we gained a series of ideas to improve this work.
>
> ---
>
> **Some miswords and confusion in the article:**
>
> We will correct all word errors in the article, such as donate  denote in lines 214, 215, dicision → decision in Fig 1. We will also use the term "preferred action" to denote the preferred action at the moment. In line 222, "traing" will be modified to "training". For weaknesses 4 and 5, we use VLM to output decision-making and COT text. So the output of VLM itself is the strategy of the agent to perform the task in the environment, We use $\pi_{ref}$ to denote the output of a VLM that has not been fine-tuned using trajectories in environments. The whole process is illustrated by "good action" in Fig. 2, where the text output of thought and action is called $\pi_\theta$. In line 186, we will try a more appropriate expression: For every planning instance, planner output a series of planning results.
>
> For Fig. 1,  the circle trying to outline in the “preferred action” and “not preferred action” represents from this time-step the thoughts and actions between two trajectories became different, so we use different colors to describe the two trajectories in Fig. 2. From this time-step, we use dpo framework to to optimize the vlm to align with the embodied environment to make decisions. Making the output of the vlm closer to the reasonable better action output at this step. So the pairs are **action level**.
>
> **For weaknesses 12**, this paper only discusses the framework of the optimized vlm under the no-reward model. As far as we know, PPO can fine-tune vlm under the framework of reward function or reward model, but often there is no built-in reward function in the environment, and it is often difficult to design the reward function manually and highly depends on experience, such as [1], so the standard adopts the preference optimization framework without reward model. We will also change our description in the contribution.
>
> ---
>
> **Some implementation details and supplementary material we will provide:**
>
> First of all, we will soon provide training details of our model in future versions of the article and in official responses to all authors. At the same time, due to the lack of careful explanation of our pipeline in our work, we will add the sft and dpo example codes in our process to the official response to all authors.
>
> ---
>
> **Some experiment details:**
>
> We use the same reward function as [1]: $r(s_t,a_t,s_{t+1}|g_{task})=50*\textbf{1} \[ s_{t+1}=g_{task}\]+\textbf{1}\[s_{t+1}=g_{sub-task}\]-\textbf{1}\[a_{t}\notin A_{adm}(s_t)\]$. From this formula, we can see that when the agents in the environment complete the subtask goal, the vlm fine-tuned in a PPO-like pipeline can be rewarded, which is also similar to providing information about the progress of the task completion. In our framework, we do not provide the signal whether the final task is completed and the signal whether the action is a legal action in the current state, so the information related to task completion obtained by our framework is less.
>
> ---
>
> [1] Large Language Models as Generalizable Policies for Embodied Tasks

---

> ### Comment · Reviewer_r5ga · 2024-11-30
>
> After reading the rebuttal, I will retain my score. The paper needs significant improvements to reach clarity in presentation. I strongly recommend that the authors incorporate into the main paper the improvements they are currently offering to add to the supplementary material.
>
> Specifically, this includes adding appropriate baselines, such as:
>
> * The direct action selection baseline, which would clearly show the advantage of the COT (Chain of Thought) setting.
> * A baseline using the same signals (e.g., task progress) for feedback. I disagree with the argument that using task progress constitutes a "less information" setting. On the contrary, it provides a much denser reward signal. Including this baseline would allow for a more rigorous comparison.

---

> > ### Author Response · Authors · 2024-12-01
> > **Clarification on Signal Usage**
> >
> > Dear Reviewer r5ga,
> >
> > Thank you very much for your valuable time and thoughtful response. We would like to further clarify the signal aspect: our baseline utilizes **sub-goal success rates**, **task completion**, and **penalties for illegal actions** as feedback signals. In contrast, when calculating the preference score, we **only considered task completion**. This means that in this step, we used less information while achieving comparable, if not better, results. We hope this response addresses some of your concerns.
> >
> > Best regards,
> >
> > Authors

---

### Official Review · Reviewer_MhUu · 2024-11-09

**Soundness:** 2
**Presentation:** 2
**Contribution:** 2
**Rating:** 5
**Confidence:** 3

**Summary:**

This paper used Direct Preference Optimization (DPO) with a Vision Language model that uses Chain-of-Thought (COT) prompting for task planning of embodied agents. The proposed method optimized the COT process instead of the final action as in PPO. The experiment showed the proposed method outperformed RL4VLM in terms of the average success rate on ALFWorld dataset.

**Strengths:**

- Replacing PPO in RL4VLM with DPO can increase average success rate.

**Weaknesses:**

- The contribution is incremental. The main contribution of this paper is replacing PPO in RL4VLM (which also has COT) [1] by DPO [2] with necessary adaptions.

- The motivation that the authors want to use DPO is that it could be beneficial in the cases of long action sequences, partial occulsion and multi-tasking. However the results do not show any of these scenarios. It's not clear whether DPO performs better thanPPO in these scenarios.

- More analyses are needed to show the performance gain on ALFWorld, details see comments below.


[1] Zhai, Yuexiang, et al. "Fine-Tuning Large Vision-Language Models as Decision-Making Agents via Reinforcement Learning." arXiv preprint arXiv:2405.10292 (2024).
[2] Rafailov, Rafael, et al. "Direct preference optimization: Your language model is secretly a reward model." Advances in Neural Information Processing Systems 36 (2024).

**Questions:**

- I'm wondering why the proposed method works much better on the task Heat & Place and Examine in Light, while worse on Pick & Place. Can authors provide more insights on these?

- What will the proposed model perform without COT prompting? Since the proposed method re-plan the task at interactive steps, is COT helpful in this case? I'm interested to see the ablation study with and without COT prompting.

- Intuitively, re-planning should work better when the action sequence is long, and the closer towards the end of the action sequence, the more accurate and less uncertain the planned actions should be. Does the proposed method behave like this?

- There are typos and grammar mistakes in the text and figure, please check the spelling.

---

> ### Author Response · Authors · 2024-11-22
>
> Thank you very much for your candid evaluation and valuable suggestions. In response to the issues you raised, we have made the following additions to certain aspects of this paper:
>
> ---
>
> **The main contribution of this paper extends beyond merely substituting the PPO component with DPO.** Our perspective emphasizes that the thoughts generated by the Chain-of-Thought (COT) output are more significant than the final actions taken. Most existing works, including RL4VLM with PPO used, primarily focus on the correctness of final decisions, which can lead to model degradation, poor generalization performance, and other issues. Therefore, our proposed **Action Probability Weighting (APW)** (see Section 3.2.2) and **Action Policy Consistency Constraints (APC)** are our key contributions. These are strongly supported by evidence presented in the ablation experiments (Sections 4.2 and 4.3). Additionally, in Equation (3), we enhance **step-wise** preference construction by modifying the original DPO optimization objective from $r(s, a)$ to $Q(s, a)$ , significantly improving sample efficiency. We will ensure that the revised version of the paper highlights these points.
>
> ---
>
> **Regarding performance on the Heat & Place and Examine in Light tasks:** Each type of task in ALFWorld requires a different average number of execution steps. Most Examine in Light tasks can be completed within a few steps (e.g., 3-4 steps). We initially infer that for simpler tasks, our method more readily acquires planning skills. We will also conduct more fine-grained case analyses in future work to further investigate this observation.
>
> ---
>
> **Concerning the ablation experiments on COT prompts:** Our work primarily investigates whether APW and APC can enhance the model’s performance. Additionally, our reference work RL4VLM has provided detailed data demonstrating that the agent's performance is significantly lower without COT. Therefore, we believe that our experimental results will remain consistent and free from unexpected outcomes. We will ensure that the revised version includes the relevant experimental data to support this.
>
> ---
>
> **Regarding whether re-planning works better when the action sequence is long:** We will incorporate experimental data addressing this aspect in the final revised version to provide comprehensive insights.
>
> ---
>
> Once again, thank you for your suggestions. In the final revised version, we will correct all wording errors and clarify the experimental details to ensure the paper’s clarity and accuracy.

---

> > ### Comment · Reviewer_MhUu · 2024-12-01
> >
> > Dear authors, thanks for the revision. However, I will keep my score.

---

### Author Response · Authors · 2024-11-27
**Looking forward to the discussion**

Dear Reviewers,

We sincerely appreciate the time and effort you've dedicated to reviewing our paper. We understand that you have a busy schedule, and we are truly grateful for your valuable feedback. As the Author-Reviewer discussion phase is coming to a close soon, we wanted to kindly reach out to see if there are any additional questions or points you'd like to discuss. We are keen to ensure that our responses have effectively addressed your concerns, and we believe that further discussion could be beneficial.

We would greatly appreciate the opportunity to engage in any further discussion. Thank you once again for your thoughtful consideration.

Best regards,

Authors

---

### Author Response · Authors · 2024-11-30
**Update the Manuscript**

Thank you once again for all your efforts! We have updated our manuscript accordingly, and the revisions are summarized as follows:

- **Writing Quality**: We made extensive revisions to enhance the clarity and readability of the paper, ensuring all implementation details are thoroughly described and included a link to the code implementation for reference.

- **Typos and Wording**: We conducted a comprehensive review of the manuscript to correct all typos and unclear wording. For example, "donate" has been changed to "denote," and the phrase in line 186 has been clarified. Additionally, "dicision" in Figure 1 is corrected to "decision," and we have clarified the relationship between "preference action" and "not preference action." All instances of $\tau_t^{t-1}$ have also been corrected to $\tau_i^{t-1}$.

- **Definition of Equations and Symbols**: All subscripts in the equations have been defined in the text for better understanding, and the typo in the section heading on line 222 has been corrected.

- **Clarification of Contribution Claims**: We clarified the contributions of the method and the role of the DPO algorithm, and we will open-source our algorithm framework shortly. We also explained the specific meaning of "tailored to the interaction characteristics of embodied agents" to eliminate any ambiguity.

- **Preference Score Definition**: A more detailed explanation of the motivation behind the preference score definition in Equation 10 has been provided, clarifying the choice of 50 * progress specifically.

- **Trajectory Collection and Score Computation**: We elaborated on the process of trajectory collection and clarified the scoring computation description to ensure understanding.

We hope that these revisions address your concern. As the discussion phase is concluding, we kindly ask you to read our response and make a reconsideration of our work. Please feel free to contact us if there are any points you'd like to discuss further. Thank you for your time and attention!

---

### Meta-Review · Area_Chair_gb78 · 2024-12-17

**Metareview:**

The paper received ratings 5,3,3, which are all below the acceptance threshold. The reviewers raised various concerns regarding missing implementation details, lack of thoroughness in the experiments, missing details required to understand the paper, incremental contribution, and the need for more analysis. The authors provided a rebuttal, but the reviewers did not find the responses convincing enough (more details below). The AC is in agreement with the reviewers and recommends rejection.

**Additional Comments On Reviewer Discussion:**

Reviewer r5ga kept the score after the rebuttal since the paper still requires major clarity improvements. They also suggested two baselines (direct action baseline and a baseline with the same signals). Reviewer MhUu was still concerned with the incremental nature of the work. The authors highlighted APW and APC as the key contributions, but the reviewer still thought the method contribution does not meet the ICLR bar. The reviewer also mentioned that the results do not show 'aligning to COT' is important. They also had concerns regarding referring to the RL4VLM paper for the ablation study of COT since RL4VLM uses PPO and it does not necessarily mean that it is the same for using DPO. They also mentioned that there is no analysis/ablations in the results, nor in the rebuttal regarding the claim that re-planning is better than the planning in RL4VLM. Reviewer fL8F also believes that significant improvement is required for the clarity of the presentation and the structure, and they want to see results beyond ALFWorld, and they believe that MSE is not appropriate to keep two categorical distributions close.

---

### Decision · Program_Chairs · 2025-01-22

Reject